# Failure strength of glacier ice inferred from Greenland crevasses

Aslak Grinsted[1], Nicholas Mossor Rathmann[1], Ruth Mottram[2], Anne Munck Solgaard[3],
Joachim Mathiesen[4], and Christine Schøtt Hvidberg[1]

[1]Physics of Ice, Climate, and Earth, Niels Bohr Institute. University of Copenhagen, Tagensvej 16, DK-2200 Copenhagen N, Denmark
[2]National Center for Climate Research, Danish Meteorological Institute, Lyngbyvej 100, Copenhagen, 2200, Denmark
[3]Geological Survey of Denmark and Greenland (GEUS), Øster Volgade 10, 1350 Copenhagen K
[4]Biocomplexity, Niels Bohr Institute, University of Copenhagen, Blegdamsvej 17. 2100 København Ø.

**Correspondence:** Aslak Grinsted (aslak@nbi.ku.dk)

**Abstract.** Ice fractures when subject to stress that exceeds the material failure strength. Previous studies have found that a von Mises failure criterion, which places a bound on the second invariant of the deviatoric stress tensor, is consistent with empirical data. Other studies have suggested that a scaling effect exists, such that larger sample specimens have a substantially lower failure strength, implying that estimating material strength from laboratory-scale experiments may be insufficient for glacier-scale modelling. In this paper, we analyze the stress conditions in crevasse onset regions to better understand the failure criterion and strength relevant for large-scale modelling. The local deviatoric stress is inferred using surface velocities and reanalysis temperatures, and crevasse onset regions are extracted from a remotely sensed crevasse density map. We project the stress state onto the failure plane spanned by Haigh–Westergaard coordinates, showing how failure depends on mode of stress. We find that existing crevasse data is consistent with a Schmidt–Ishlinsky failure criterion that places a bound on the absolute value of the maximal principal deviatoric stress, estimated to be $(158 \pm 44)\,\mathrm{kPa}$. Although the traditional von Mises failure criterion also provides an adequate fit to the data with a von Mises strength of $(265 \pm 73)\,\mathrm{kPa}$, it depends only on stress magnitude and is indifferent to the specific stress state, unlike Schmidt–Ishlinsky failure which has a larger shear failure strength compared to tensile strength. Implications for large-scale ice-flow and fracture modelling are discussed.

## 1 Introduction

Understanding the mechanics of ice fracture is important for predicting the stability of ice sheets and glaciers, since fractures lead to crevassing, calving, and moulins (Colgan et al., 2016): calving is a major component of the mass budget of ice sheets and a potential source of rapid unstable retreat; crevassed and damaged ice is expected to flow more readily (Borstad et al., 2013; MacAyeal et al., 1986; Albrecht and Levermann, 2014; Sun et al., 2017); moulins couple basal conditions to surface conditions (Das et al., 2008) by channeling warmer surface melt water through the englacial system and potentially accelerating rates of ice loss (cryohydrologic warming; e.g. Phillips et al. (2010); Solgaard et al. (2022)). Yet in spite of the broad literature on glacier ice, important material properties that constrain ice fracturing remain under-studied.

In classical failure theory, fractures form when an appropriate measure of the internal stress magnitude exceeds a critical value (material failure strength). Fracturing leads to a rapid elastic response with a redistribution of the internal stress, tending

to concentrate near material defects and crack tips. Fracture propagation, and, therefore, ultimately the size of crevasses, is controlled by how the local stress field evolves in the presence of a crack. This kind of propagation has previously been modelled using linear elastic fracture mechanics that relies on material parameters such as the fracture toughness (van der Veen, 1998; Petrovic, 2003; Albrecht and Levermann, 2014).

There is, however, considerable discrepancy between failure strength estimates derived from lab experiments (Hawkes and Mellor, 1972; Haynes, 1978; Petrovic, 2003) and field data (Kehle, 1964; Ambach, 1968; Vaughan, 1993; Petrovic, 2003). Lab experiments found the tensile strength of ice to be $1.7\,\text{MPa}$ to $3.2\,\text{MPa}$ for temperatures above $-37\,°\text{C}$, whereas estimates inferred from field measurements are an order of magnitude smaller, $90\,\text{kPa}$ to $320\,\text{kPa}$ (Vaughan, 1993). While such difference could be partly due to material differences like crystal fabric, grain size distribution, or impurity content, it is more likely the result of sample size affecting failure strength: larger samples have substantially reduced tensile strength compared to smaller samples (Dempsey, 1996; Dempsey et al., 1999) which has been attributed to how the probability of weakest-links scale with volume (Petrovic, 2003). Specifically, a factor 10 increase in the characteristic length scale has been found to reduce the apparent tensile strength by a factor three (Dempsey et al., 1999).

Several failure criteria have been proposed in the literature. Vaughan (1993) used failure maps to justify that both Coulomb and von Mises failure criteria conform well with empirical crevasse data. Lab experiments have meanwhile found that the compressive strength is an order of magnitude greater than the tensile strength (Haynes, 1978; Petrovic, 2003), which is in apparent contradiction to von Mises failure theory. Furthermore, the von Mises criterion is independent of pressure, but Nadreau and Michel (1986) found that failure stresses increase with hydrostatic pressure.

In summary, there is a discrepancy between the failure strength inferred from laboratory and field observations, and multiple distinct failure criteria have been proposed. In this paper, we use a large data set of Greenland crevasses to estimate a macro-scale failure criterion for naturally occurring glacier ice, relevant for large-scale modelling.

## 2 Data

Chudley et al. (2021) derived a Greenland Ice Sheet wide crevasse map by processing high resolution elevation data from the ArcticDEM v3 mosaic (Porter et al., 2022), which in turn is based on remote sensing data collected over the period from 2007-2015. In this paper, we use the 200m resolution crevasse density data set (fig. 1) which contains the area fraction of 2x2m pixels that has been classified as crevassed (Chudley, 2022). This optically-derived data set is insensitive to snow-filled crevasses and it therefore likely underestimates the crevasse extent at higher elevations (Chudley et al., 2021).

Strain rates are derived from ice velocity measurements from the Greenland Ice Sheet Velocity Mosaic (Joughin et al., 2016, 2018) which is a multi-mission velocity average spanning the years from 1995 to 2015 (fig. 2). Surface air temperatures at 2.5km resolution are taken from the CARRA reanalysis data (Schyberg et al., 2021) averaged over the period from 1991-2020 (fig. 2).

We compare the calculated von Mises stress to BedMachine v5 ice thickness ($H$) (Morlighem, 2022), ArcticDEM ice sheet elevation ($z$) (Porter et al., 2022), along flow acceleration since 1985 ($\dot{v}$) (Grinsted et al., 2022; Grinsted, 2022), and the

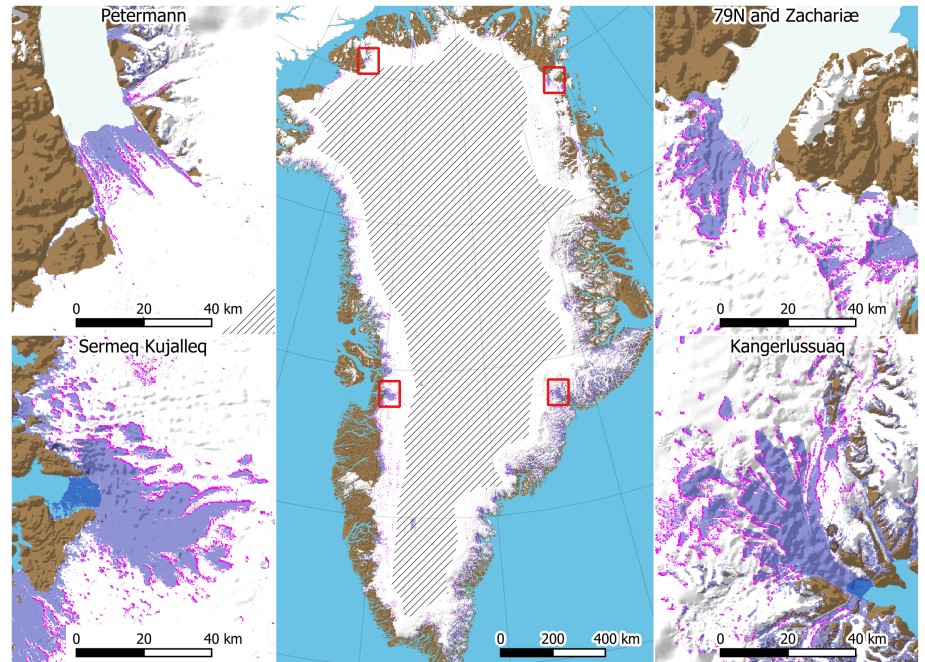

**Figure 1.** The spatial distribution of crevasses in Greenland from Chudley et al. (2021) is shown in semitransparent lavender blue. The crevasse onset regions along the upstream edge of crevasse fields are shown in magenta. The hatched region in the interior of the ice sheet has been excluded from the analysis.

seasonal amplitude in ice velocities ($V_{\mathrm{peak}}/V_{\mathrm{winter}}$) derived from PROMICE Sentinel-1 ice velocities (Solgaard and Kusk, 2021; Solgaard et al., 2021).

## 3   Methods

Our aim is to relate crevassing to the local stress environment where crevasses initiate. Evidence suggests that most crevasses initiate at a depth of 15-30m (Colgan et al., 2016), which in a Greenland context can be considered near surface. We will therefore assume that the horizontal velocities at initiation depth are equal to surface velocities. We furthermore assume that vertical shear components are neglible and write the strain rate tensor as

$$\dot{\boldsymbol{\epsilon}} = \begin{pmatrix} \dot{\epsilon}_{xx} & \dot{\epsilon}_{xy} & 0 \\ \dot{\epsilon}_{xy} & \dot{\epsilon}_{yy} & 0 \\ 0 & 0 & \dot{\epsilon}_{zz} \end{pmatrix}, \tag{1}$$

where the horizontal components ($\dot{\epsilon}_{xx}$, $\dot{\epsilon}_{xy}$, $\dot{\epsilon}_{yy}$) are calculated from the observed surface velocities (Joughin et al., 2016). Crevasses tend to form at lower elevations where temperatures are warmer and firn densities increase more quickly with depth. Theoretical arguments suggest that crevasses initiate below the firn as greater stresses are possible in ice (van der

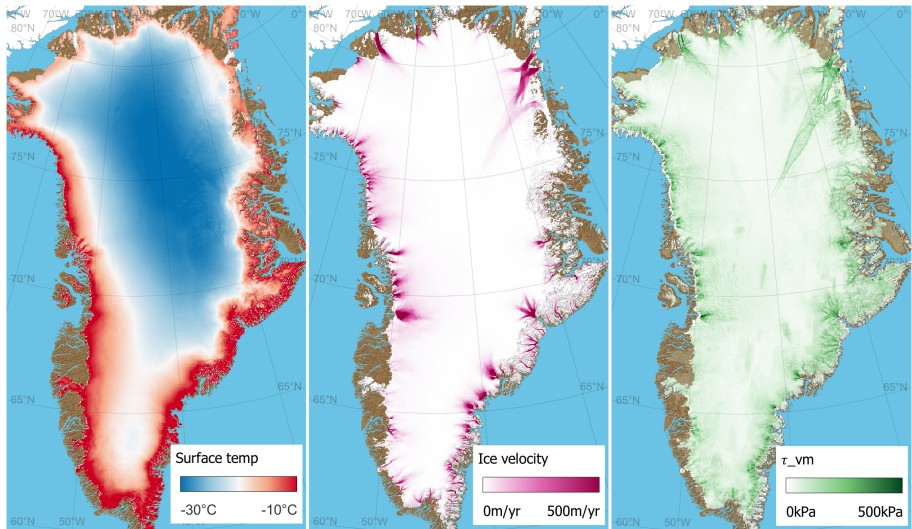

**Figure 2.** Average surface air temperatures from CARRA (left), Ice velocities from MEAsUREs (middle), and near surface von Mises stress calculated assuming solid ice rheology (right).

Veen, 1998). Indeed, this is supported by observations of crevasses initiating from refrozen melt layers (Scott et al., 2010; Christoffersen et al., 2018). We therefore assume that fractures initiate in the solid ice and hence that incompressibility applies, $\dot{\epsilon}_{zz} = -\dot{\epsilon}_{xx} - \dot{\epsilon}_{yy}$. Finally, we disregard the effect of pressure on the failure envelope (Nadreau and Michel, 1986) following Vaughan (1993) since only surface crevasses are considered here; that is, crevasse formation subject to relatively small ice pressure.

Glen's flow law for solid ice relates the strain-rate tensor to the deviatoric stress tensor ($\tau_{ij}$) as

$$\dot{\epsilon}_{ij} = A(T)\tau_{\mathrm{e}}^{n-1}\tau_{ij}, \tag{2}$$

where $A$ is a temperature-dependent rate factor, $n$ is the flow exponent, $\tau_{\mathrm{e}} = \sqrt{I_2} = \sqrt{\tau_{ij}\tau_{ij}/2}$ is the effective deviatoric stress, and $I_2$ is the second invariant of the deviatoric stress tensor. We calculate $\tau_{\mathrm{e}}$ from the effective shear strain rate $\dot{\epsilon}_{\mathrm{e}}$ using $\tau_{\mathrm{e}} = [\dot{\epsilon}_{\mathrm{e}}/A(T)]^{1/n}$. Here, we take the canonical value for the flow exponent, $n = 3$, and set the rate factor following Cuffey and Paterson (2010). For simplicity we assume that the temperature at the crevasse initiation depth can be approximated by the CARRA annual average surface air temperature. With these assumptions, we can calculate the deviatoric stress tensor and the corresponding von Mises stress

$$\tau_{\mathrm{vM}} = \sqrt{3I_2} = \sqrt{\frac{(\tau_1 - \tau_2)^2 + (\tau_2 - \tau_3)^2 + (\tau_3 - \tau_1)^2}{2}}, \tag{3}$$

where $\tau_1$, $\tau_2$ and $\tau_3$ are the principal deviatoric stresses.

## 3.1 π plane

The stress conditions where failure occurs can be represented by a three-dimensional threshold surface, spanned by the three principal stress components. For pressure-insensitive materials, the threshold surface must be reflection symmetric along the hydrostatic axis ($\sigma_1 = \sigma_2 = \sigma_3$). This symmetry implies that the threshold surface can, without ambiguity, conveniently be represented on a two-dimensional subspace (plane) by a threshold curve (or 'failure envelope'). A popular choice is the so-called $\pi$-plane view which corresponds to a plane perpendicular to the hydrostatic axis.

In the following, we will determine the failure criteria of ice by plotting principal deviatoric stresses in the $\pi$-plane and search for a match with hypothesized polygon profiles representing different failure criteria (Kolupaev, 2018). The $\pi$-plane is the plane spanned by the Haigh–Westergaard coordinates $\xi_2$ and $\xi_3$, related to the principal deviatoric stresses through the transformation

$$\xi_2 = \frac{1}{\sqrt{2}}(\tau_1 - \tau_3) \tag{4}$$

$$\xi_3 = \frac{1}{\sqrt{6}}(-\tau_1 + 2\tau_2 - \tau_3). \tag{5}$$

In this plane, the von Mises failure criterion appears as a circle with radius $r = \sqrt{2/3}\tau_{vM}$ since

$$\xi_2^2 + \xi_3^2 = \frac{(\tau_1 - \tau_2)^2 + (\tau_2 - \tau_3)^2 + (\tau_3 - \tau_1)^2}{3} = \frac{2}{3}\tau_{vM}^2, \tag{6}$$

where $\tau_{vM}$ is the material failure strength (critical stress).

An alternative visualization (Kolupaev, 2018, ch3.4), common in the glaciological literature, is to focus on the plane where $\sigma_3 = 0$ (i.e. plot $\sigma_1$ vs $\sigma_2$) (Vaughan, 1993, e.g.). In this plane, the von Mises envelope appears as an ellipse tilted 45 degrees. However, Glen's flow law is concerned with $\tau_{ij}$, not $\sigma_{ij}$, so it is common to plot $\tau_1$ versus $\tau_2$ instead, calculated from horizontal velocities (Chudley et al., 2021, e.g.). In this case, results must be considered carefully to avoid misinterpretations: the approach implicitly assumes that $\tau_3 = 0$, which is not valid for horizontally diverging flows. If $\tau_3 = 0$ is not fulfilled we cannot expect points to fall on a von Mises ellipse even if generated by a von Mises failure process. We therefore advocate for the $\pi$-plane visualization as it is independent of pressure (i.e. plots for $\sigma$ or $\tau$ gives the same result) and, importantly, takes all principal stress components into consideration.

## 3.2 Regions of interest

Crevasses are transported with the flow, and so not all crevasses are a product of the local stress conditions. As we are interested in the failure strength, we create a mask to select locations where crevasses have recently opened. These are defined as being located on the uphill edge of large-scale crevasse patches. We therefore apply a smoothed Sobel edge detection filter to a binary representation of the crevasse density map, and select the upstream edge by requiring that the along-flow crevasse density gradient to increase. We further require the crevasse density to be smaller than 5%. We label these as *crevassing onset* regions.

The high-elevation ice sheet interior has been excluded from the analysis, using a manually traced polygon (Grinsted, 2024), as we gauge that the false positive rate in the crevasse product is greatest there (see fig. 1). Regions where the ice thickness is
below $200\,\mathrm{m}$ are also excluded due to concerns of whether the velocity product has sufficient spatial resolution to resolve the local strain rate.

Strain rates and stresses are calculated from long-term average ice velocities. This may not accurately reflect the conditions under which crevasses were formed since trends or seasonal variability in ice flow are unaccounted for (Grinsted et al., 2022; Solgaard et al., 2022) (elaborated on below). We therefore separately examine onset regions with *steady flow*, defined as
fulfilling $V_{\mathrm{peak}}/V_{\mathrm{winter}} < 2$ and $\dot{v} < 2\,\mathrm{m\,a^{-2}}$.

## 4 Results

The spatial distribution of crevasses and the automatic detection of *crevasse onset* regions are shown in fig. 1. Maps over the calculated von Mises stress, the ice velocity, and the surface temperature are shown in fig. 2. Calculated von Mises stresses are found to be greater in onset regions compared to crevassed regions in general, and over the ice sheet as a whole (fig. 3). Onset
regions with steady flow are characterized by even greater von Mises stresses, $\tau_{\mathrm{vM}} = (265 \pm 73)\,\mathrm{kPa}$ (fig. 3). The regional differences in the von Mises stress distribution is shown in the appendix (fig. A1). The distribution of principal stresses in onset regions with steady flow are shown in figure 4. A similar plot for the distribution of strain rates is shown in fig. B1. In fig. A2, we show how the von Mises stress in onset regions relates to other field variables.

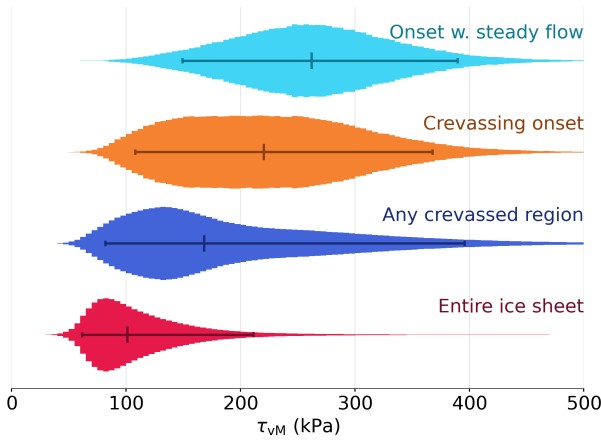

**Figure 3.** The von Mises stress distribution over different subsets of the ice sheet. The counts per bin are all normalized to have the same peak height. Horizontal bars show the 5%, 50% and 95% percentiles of the distribution.

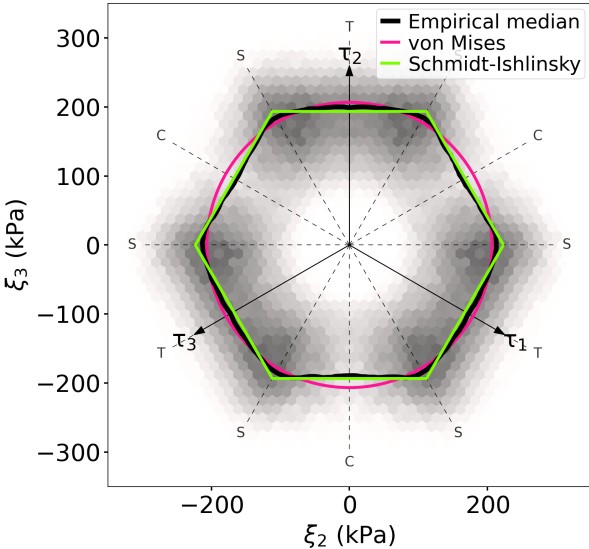

**Figure 4.** Empirical failure map showing a $\pi$-plane density map of stresses in crevasse onset regions with steady flow. The grey color scale is linear in the counts per bin. The empirical median in $5°$ windows is shown in black; The median von Mises stress is shown as a pink circle; And the Schmidt–Ishlinsky median maximum absolute deviatoric stress is shown as a green hexagon. Tensile, Compressive and Shear directions have been labelled with T, C, and S.

## 5 Discussion

### 5.1 Uncertainties

Stresses are calculated assuming that ice temperatures, at the depth of crevasse initiation, can be approximated by surface air temperature as simulated by the CARRA regional reanalysis (fig. 2). It is, however, not clear whether this model is accurate, and there will likely be regional biases in the modelled temperature. Further, ice temperatures at depth may deviate from surface temperatures due to advection and redistribution of energy via melt and refreezing (Løkkegaard et al., 2022; Harrington et al., 2015). We gauge the sensitivity of our stress estimates to temperature by calculating $(A(T_1)/A(T_2))^{1/n}$ (see eqn. 2), suggesting that a $1°C$ change in temperature may result in inferred deviatoric stresses changing by $4\%$ to $8\%$. Note that canonical parameters for Glen's flow law were here assumed that disregard the effect of impurities or crystal fabric on ice rheology.

We find that crevasse onset regions are characterized by larger stresses than in crevasse fields in general (fig. 3). This is not surprising as many crevasses have been transported with the flow after being formed upstream. It is therefore clear that the von Mises stress distribution in crevasse fields does not, in general, accurately reflect the failure stress. We moreover find substantial regional differences in the onset von Mises stress distributions (fig. A1). Ideally, the estimated failure stress should be independent of region, so this suggests that we are not accurately estimating the in-situ stress everywhere. We look

for clues that might explain regional discrepancies by examining the how the onset region stress co-varies with a range of local conditions (fig. A2), finding a strong anti-correlation between the von Mises stress and the seasonal amplitude in ice velocities (Rank correlation of $-0.59$). We interpret this as our method systematically underestimating the failure stress in regions with strong seasonal variations in near surface strain rates, and therefore argue that onset regions with steady flow more accurately represent the failure stress. Excluding regions with non-steady flow results in a narrower stress distribution that has an improved separation from the stress distribution over ice sheet as a whole (fig. 3). We therefore estimate the failure stress to be $\tau_{vM} = (265 \pm 73)\,\mathrm{kPa}$ from onset regions with steady flow. This separation is not as clear in Chudley et al. (2021) which is based on a regional analysis of the same crevasse dataset but does not exclude areas with non-steady flow.

We note that not every location where estimated stresses exceed the failure stress appear to be crevassed (fig. 2). This may be due to limitations in the crevasse dataset which is insensitive to snow-filled crevasses, or because exceeding the failure criterion is a necessary but not sufficient condition for crevasse formation – conditions must also favour crack *propagation*. Furthermore, in some interior locations, cracks may initiate in the firn pack, thus requiring a failure criterion for firn and a compressible rheology (e.g. Gagliardini and Meyssonnier, 1997; Petrovic, 2003).

## 5.2 Failure criterion

The empirical failure map in fig. 4 shows that in steady onset regions, crevasses predominantly open when subject to large shear or tension, and much less so compression. This does not necessarily reflect the relative failure strength of the different modes, but may simply reflect that the greatest surface stresses occur in shear zones. We therefore estimate the empirical failure envelope as the median of the data in $5°$ windows in the $\pi$-plane (Solid black line in fig. 4). The empirical envelope shows that ice in tensile and compressive deformation is nearly equally strong, with the tensile failure strength only being 4% greater than the compressive strength (see fig. 4). The empirical envelope compares reasonably well with the von Mises failure criterion which appears as a circle in the $\pi$-plane. However, the data clearly has a hexagonal pattern which we compared against a library of common failure criteria (Kolupaev, 2018, ch.3). The hexagonal shape, with corners in the shear directions, strongly suggests that glacier ice follows a Schmidt–Ishlinsky failure criterion (Burzynski, 1928; Schmidt, 1932; Yu, 1983; Kolupaev, 2018).

In the Schmidt–Ishlinsky hypothesis, it is the largest principal deviatoric stress, rather than the magnitude of the second invariant, that defines the failure criterion:

$$\tau_{SI} = \max(|\tau_1|, |\tau_2|, |\tau_3|), \tag{7}$$

where the material failure strength (critical stress) is derived from the inscribed radius $r$ of the hexagon profile in the $\pi$-plane using $\tau_{SI} = \sqrt{2/3}\,r = (158 \pm 44)\,\mathrm{kPa}$. The exscribed radius, corresponding to the shear failure strength is given by $R = \sqrt{4/3}\,r$, i.e. the Schmidt–Ishlinsky criterion implies that ice is 15% stronger in shear relative to tensile stresses. Other polycrystalline materials—such as mild steel, copper, nickel alloy, titanium, stainless steel—have been found to have Schmidt-Ishlinsky like behaviour (see Kolupaev, 2018, table 2.1).

Alternative criteria have also been proposed in the literature, such as the Coulomb criterion and the maximum strain-energy dissipation criterion (MacAyeal et al., 1986; Vaughan, 1993), but are inconsistent with our data as they imply that ice is

weakest for shear. We quantify the misfit between previously-proposed theoretical failure criteria and the empirical envelope by calculating the root mean square log deviation (logRMS). Glen's flow law allows for only deviatoric stress to be estimated, so we are prevented from considering (calculating misfits of) pressure-dependent criteria. The pressure invariant version of the Coulomb criterion is known as the Tresca criterion. We find that the Schmidt–Ishlinsky fits the data best with a logRMS error of 0.02. For comparison, the von Mises and Tresca criteria have logRMS errors of 0.04, and 0.08 respectively. Further, measurement noise acts as to blur the octahedral failure profile, thus softening the corners of the hexagonal profile. We therefore argue that the Schmidt–Ishlinsky criterion is an even better failure model for glacier ice than it appears from the empirical envelope.

In the maximum strain energy dissipation criterion (Vaughan, 1993), it is the rate of deformational work ($P = \dot{\epsilon}_{ij}\tau_{ji}/2$) which is hypothesized to be limited, rather than the stress state that can be withstood by the material. Vaughan (1993) uses Glens flow law (eqn. 2) to calculate $\tau$ corresponding to a $\dot{\epsilon}$. Unfortunately, estimating the $\tau$ failure envelope corresponding to a given threshold value for $P$ becomes ambiguous as there will be a separate failure curve for every temperature. We can therefore not evaluate the fit using the same logRMS metric we used for the other criteria. However, we calculate the standard deviation of $\log(P)$ to be 1.2 in crevasse onset regions with steady flow. This large spread indicates that there is not a single threshold value for $P$, and we therefore have little confidence in relevance of the maximum strain energy dissipation criterion. For comparison, the log-standard deviations of the estimated threshold values for the von Mises and the Schmidt–Ishlinsky criteria, both have log-standard deviations of 0.3, and thus are much better fits to the data.

We find that ice thickness correlates positively with the von Mises stress (fig. A2), contrary to expectations based on the established volume scaling effect. We speculate that the failure strength might approach a limiting value for large ice sample sizes as predicted by e.g. the multi-fractal scaling law (Carpinteri et al., 1995; Dempsey et al., 1999). This would imply that our reported critical stress values can be used in large-scale ice sheet models without adjustment for the scale effect.

### 5.3 Modelling crevasse fields

The von Mises and Schmidt–Ishlinsky failure envelopes do not deviate strongly from each other, but, for a given stress magnitude, the failure criterion might be fulfilled in one case and not the other depending on the stress state: the von Mises criterion is indifferent to the stress state, whereas the Schmidt–Ishlinsky criterion favors failure by tension or compression for a given stress magnitude; that is, glacier ice appears stronger when subject to shear as opposed to tensile stresses (see fig. 4).

The choice of failure criterion may therefore impact where crevasses are formed in large-scale models. A popular approach is to model the evolution of the crevasse density field (or damage phase field) as a function of local conditions and couple it back into ice viscosity (e.g. Albrecht and Levermann, 2012, 2014; Borstad et al., 2016; Sun et al., 2017; Kachuck et al., 2022). In such models, crevasse density evolution is represented as a production–healing process, where production depends on a scalar measure of the local stress state, which, if exceeding some critical value, is taken to grow proportionally to the local principal spreading rate (strain rate). Our results are therefore directly applicable to such modelling efforts: we provide both the failure criterion (7) and critical value $\tau_{\mathrm{SI}} = (158 \pm 44)\,\mathrm{kPa}$. We leave it for future work to test whether modelled crevasse density fields using the Schmidt–Ishlinsky criterion in fact provide the best fit with observations.

## 5.4 Third stress-tensor invariant

The Schmidt–Ishlinsky criterion implies that the failure of ice depends not only on the second, but also the third invariant of the deviatoric stress, $I_3 = \tau_{ij}\tau_{jk}\tau_{ki}/3$, since eqn. (7) can equivalently be written as (Yu, 1983)

$$\left(I_3 + I_2\tau_{\mathrm{SI}} - \tau_{\mathrm{SI}}^2\right)\left(I_3 - I_2\tau_{\mathrm{SI}} + \tau_{\mathrm{SI}}^2\right) = 0. \tag{8}$$

The plastic behaviour and failure of materials are often linked, which suggests that the third invariant should be included in the flow law for ice; a dependency that has previously been proposed (Glen, 1958; Veen and Whillans, 1990; Morland and Staroszczyk, 2019; Baker, 1987). Indeed, Steinemann (1954) found that ice deforms slower in shear than expected from uniaxial deformation tests (Glen, 1958), and suggested that deformation data are not consistent with Glen's canonical co-axial flow law with a dependence on only $I_2$, but should depend on $I_3$, too. Although this seems to be consistent with the Schmidt–

Ishlinsky failure pattern found here (fig. 4), constraining the stress response functions in a flow law depending on both $I_2$ and $I_3$ has proven difficult from available data (Morland and Staroszczyk, 2019; Staroszczyk and Morland, 2022). On that note, Staroszczyk and Morland (2022) recently showed that a quadratic (non-coaxial) flow law depending only on $I_2$ can also be constructed to improve the fit with deformation experiments. Nonetheless, we believe it is worth considering whether the failure criterion might, too, inform on the form of the flow law in addition to correlating with deformation data.

## 6 Conclusions

We automatically identified crevasse onset regions from a dataset of Greenland crevasses (fig. 1), where we argue local stress conditions must have exceeded the failure stress. We inferred the local stress conditions using Glen's flow law combined with observed ice velocities and average surface air temperatures (fig. 2), disregarding regions with seasonally variable ice flow. We estimated the failure strength of glacier ice to be $\tau_{\mathrm{vM}} = (265 \pm 73)\,\mathrm{kPa}$ in crevasse onset regions with steady flow (fig. 3). This

is compatible with the $90\,\mathrm{kPa}$ to $320\,\mathrm{kPa}$ tensile strength estimated by Vaughan (1993). The corresponding $\pi$-plane failure map (fig. 4) suggests, however, that the mechanical failure of glacier ice is best modelled using the Schmidt–Ishlinsky failure criterion, where failure occurs once the maximum absolute deviatoric stress exceeds $(158 \pm 44)\,\mathrm{kPa}$. While we argue that the a Schmidt–Ishlinsky criterion is the better model for ice failure, we note that the deviations to the von Mises criterion are quite small (fig. 4). It may therefore still be a good approximation to model ice failure with a von Mises criterion on average. This,

however, disregards the effect that the mode of deformation has on the failure strength, where the Schmidt–Ishlinsky criterion implies that ice is 15% stronger in shear relative to tensile stresses.

    This study is, to our knowledge, the first to propose a Schmidt–Ishlinsky failure criterion for glacier ice. More work is needed to validate this hypothesis using e.g. forward modelling.

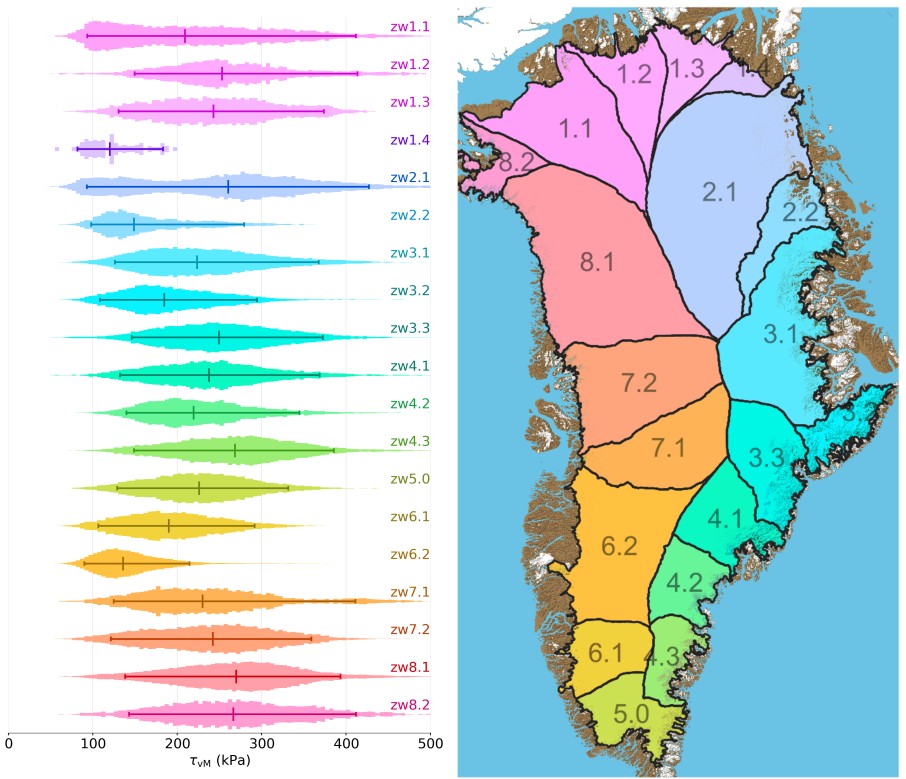

**Figure A1.** The calculated von Mises stress for opening crevasses varies substantially between different basins of the ice sheet. The counts per bin are all normalized to have the same peak height. Horizontal bars show the 5%-50%-95% percentiles of the distribution.

## Appendix A: Regional von Mises stress

The failure strength of ice should largely be independent of location. We therefore plot the onset von Mises stress distribution by region (Zwally et al., 2012). We find that there is substantial variability between regions (fig. A1). To illuminate the source of the scatter we examine how the onset von Mises stress varies with other fields in a pair plot (fig. A2. We transform variables which have a long tailed distributions with either a logarithm or hyperbolic tangent function prior to visualization. We find that the von Mises stress anti-correlates with seasonal velocity amplitudes. This suggests that the von Mises stress, calculated from

long-term average velocities, are biased low in some regions as it does not capture the seasonal peak stress which may drive crevasse formation.

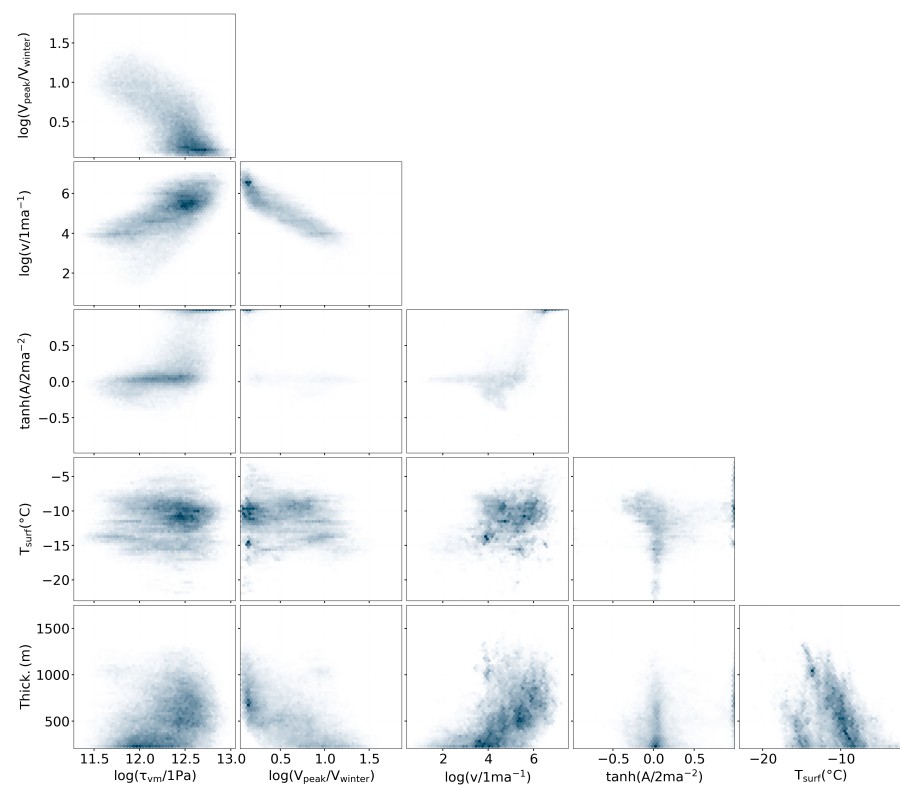

**Figure A2.** Pair plot showing the relationships between various quantities extracted over regions where crevasses are opening. The variables plotted are: the von Mises stress ($\tau_{vM}$); the seasonal peak velocity relative to the winter velocity ($V_{peak}/V_{winter}$); the mean velocity from MEAsUREs ($v$); the long term acceleration in the along-flow direction ($A$); the surface temperature ($T_{surf}$); and the ice thickness (Thick.). Some variables with long-tailed distributions have been transformed using logarithms or hyperbolic tangent.

## Appendix B: Critical strain rates

In this paper, we focus on the stress conditions that result in failure of glacier ice. We do not observe the stress state, but infer it from observed strain rates and our understanding of the mechanical properties of ice. In this section, we present the raw strain rate data. To investigate the how the critical strain rate depends on the mode of deformation we define strain rate $\pi$-plane coordinates in an equivalent way to eqn. 4, but using the three invariants of $\dot{\epsilon}_{ij}$ rather than $\tau_{ij}$. Figure B1 shows the observed strain rates in crevasse onset regions with steady flow. Vaughan (1993) found that there is a strong impact on temperature on the critical strain rate, which can largely be accounted for by the rate factor in Glen's flow law (eqn. 2). We therefore calculate empirical failures envelopes in three different temperature intervals (fig. B1), and find that the critical strain rate varies by an order of magnitude between them. This is reflected in a long-tailed strain rate distribution. All three empirical failure envelopes show a similar flower-like pattern where the critical strain rate is greater for shear than for tensile and compressive deformation.

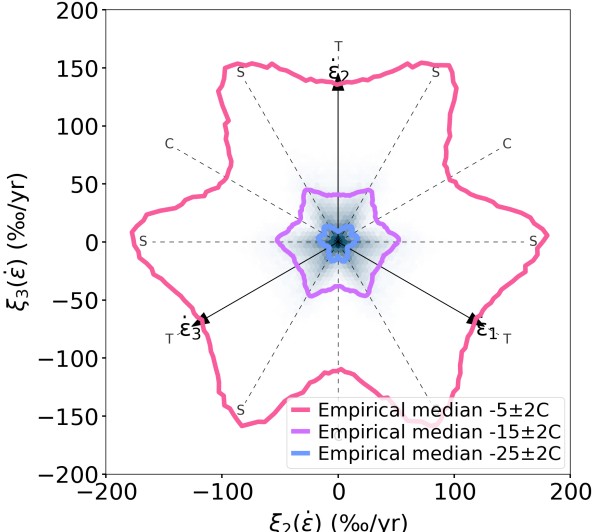

**Figure B1.** Empirical failure map showing a $\pi$-plane density map of strain rates in crevasse onset regions with steady flow. $\pi$-plane coordinates are calculated from strain rates rather than deviatoric stresses. The color scale is linear in the counts per bin. The empirical median in $10°$ windows for different temperature intervals is shown in thick lines; Tensile, Compressive and Shear directions have been labelled with T, C, and S.

*Data availability.* Crevasse density data are available from Chudley (2022). PROMICE ice velocity data from Sentinel-1 is available from Solgaard and Kusk (2021). MEaSUREs Multi-year Greenland Ice Sheet Velocity are available from Joughin et al. (2016). flow acceleration data are available from Grinsted (2022). CARRA temperatures are available from Schyberg et al. (2021). BedMachine v5 ice sheet geometry are available from Morlighem (2022). Supplemental data to this study containing masks and derived data are available from Grinsted (2024).

*Author contributions.* A.G. conceived the study, performed computations, and wrote the first draft. A.S. provided seasonal velocity amplitudes. All authors discussed the results and contributed to the final manuscript.

*Competing interests.* Some authors are members of the editorial board of The Cryosphere.

*Acknowledgements.* The research leading to these results has received funding from the Villum Foundation, investigator grant nos. 16572 and 23261, the Independent Research Fund Denmark (DFF), grant no. 2032-00364B, and it is supported by the Novo Nordisk Foundation Challenge grant no. NNF23OC0081251 .

Ice velocity maps were produced as part of the Programme for Monitoring of the Greenland Ice Sheet (PROMICE) using Copernicus Sentinel-1 SAR images distributed by ESA, and were provided by the Geological Survey of Denmark and Greenland (GEUS) at http://www.promice.dk.

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
