# Peer review of "Failure strength of glacier ice inferred from Greenland crevasses"

_EGUsphere, 2023_

## Referee Comment (RC2)

**Reviewer comments on "Failure strength of glacier ice inferred from Greenland crevasses" by Grinsted et al.**

This is an excellent paper, addressing an under-studied but important issue in glaciology in an elegant and rigorous manner. The methods and results are presented efficiently and clearly, with enough detail to address the important issues but without clutter or un-necessary material. It is pleasing to see such clear patterns emerge from the data, despite the many potential issues with data resolution.

My only substantial criticism is that the crevasse onset criteria should also be presented in terms of strain rates, rather than stress metrics alone. As the authors clearly state, the calculated stresses depend on the choice of rheology. Standard values have been used, and the prefactor $A$ has been scaled to temperature; this is all good, and aligns with standard practice in glaciology. However, major sources of uncertainty remain, including the true temperatures at crevasse-initiation depth, non-temperature influences on $A$, and the possibility (indeed, likelihood) of varying $n$ across the very large study area. For these reasons, the calculated stresses are not absolute, but parameter-dependent. The authors have done an excellent job of highlighting these issues in the text, and I have no issue with that. However, it would be very useful to present the raw strain rate values – these are the *observations*, and are hence free from any assumptions regarding the rheology. Including the strain rate data will offer researchers greater flexibility in how they interpret and use the results presented in this paper. I do not see any need to adjust what is already written in the paper, simply to add a section (and a Figure) on the strain rates.

I found only one typo. On line 82, one 'principle' sneaked into the text. As is the case elsewhere in the paper, this should of course be 'principal'.

The author team are to be congratulated on a fine study. The paper is likely to be widely cited: I shall certainly find it very useful for my work on the role of crevasses in calving.

Doug Benn

---

## Author Comment (AC2)

Responses to Reviewers' Comments for Manuscript https://doi.org/10.5194/egusphere-2023-1957

**Failure strength of glacier ice inferred from Greenland crevasses**

Addressed Comments for Publication to
EGUSphere (The Cryosphere)

by
A. Grinsted, N. Rathmann, R. Mottram, A. M. Solgaard, J. Mathiesen, and C.S. Hvidberg

**Response to RC1**

> **General Comments.** Grinsted et al. use ice-sheet-wide geophysical datasets to assess the applicability of a previously untested failure criterion, the Schmidt-Ishlinsky (hereafter S-I) criterion, to predict ice failure across the Greenland Ice Sheet. This is done with the intention of being able to develop accurate predictors of crevasse formation and presence in large-scale ice-flow and fracture models. The paper is timely and is a clear evolution of the ongoing large-scale work being done on ice failure in recent years. It is a bit of an "open secret" that von Mises (hereafter vM) is a poor predictor of crevasse failure and it is good to see work being done to find better approaches, especially involving large-scale datasets. I have some thoughts regarding the failure criterion chosen, and in particular the lack of wider assessment given the prime opportunity provided by the datasets collated.

**Response:** Thank you for your feedback.

From our perspective we would not say that we assess the applicability of the Schmidt-Ishlinsky criterion, as it is not a criterion we propose a priori. Rather we make an empirical study of the failure criterion, and then find that it happens to be well modelled by the S-I criterion.

We hope that we address your detailed comments in our replies below.

> **Comment 1**
>
> As written, it is unclear what motivation or hypothesis led the authors to propose and test the S-I failure criterion. After the abstract, the criterion is not mentioned again until the discussion (L136), where it is defined but without any clear motivation as to why. At this point, it is argued that although both vM and S-I perform quantitatively identically (L144), if measurement noise was better S-I would likely be the better performer (L144-146). It is not possible for the reader to assess whether S-I performs better or worse than any other alternative criteria common in the literature, as this data is 'not shown' (L147-148).

15 **Response:** The motivation for the S-I criterion is purely empirical - It is *the only* simple failure criteria that fits the data (simple in the sense that it has no shape parameters). We realize that this may not have been clear as S-I is introduced in the methods before any results are shown.

Here's how we arrived at that conclusion. We calculate an empirical failure envelope by calculating the moving 5°median radial distance in the $\pi$-plane. This is shown as a fat solid black line in figure 4. It has a distinct hexagonal shape that we 20 can compare that against a library of failure envelopes (see fig3.6 and fig3.7 in Kolupaev, 2018). We judged that this was so distinctly Schmidt-Ishlinsky type that we not need to quantitatively assess the misfit of different criteria. However, we have since done so, and find that the root mean square log misfits of the S-I, vM, and Tresca envelopes are 0.02, 0.04 and 0.08 respectively. Hence, we can now also conclude that the S-I criterion objectively provides the best fits to the data.
* * *
In our revisions we intend to make the following changes:

- Make it 100% clear how we arrived at the S-I criterion.

- Expand on the description of how we calculated the empirical failure envelope.

- Score the different failure criteria in terms of a RMS misfit between the proposed failure envelopes and the empirical failure envelope.

- Include the misfit score of additional (worse) failure criteria in the literature such as the Tresca/Mohr-Coulomb.

We will probably not plot additional failure criteria in figure 4, as it is already hard to distinguish the three lines.
* * *
**Comment 2**

The S-I is not a criterion I have encountered before in the glaciological literature, and appears to be pretty niche outside the discipline as well - indeed, googling the phrase 'Schmidt-Ishlinsky failure criterion' includes this preprint among the top results. As a result, I think it is important that the authors do more to contextualise and explain the criterion, and their motivation for choosing it. This is especially true as the authors disregard even comparing the criterion to other options, as previously noted (L147-148). Given the clear effort that has been put into producing and collating the various ice-sheet-wide datasets, it would be very interesting to see a comparison/EDA of all previously suggested criteria, and make it more convincing that the S-I criterion is (qualitatively) a better option.

25 **Response:** The motivation for the criterion is entirely empirical. It simply fits the data best (see our response to comment 1).

The difference between the vM criterion and S-I is relatively subtle considering the imperfections of real world data. So, it is not that surprising to us that the S-I like behaviour of glacier ice has not been spotted before. It may seem like an exotic criterion, but consider that nearly the entire ice flow literature uses Glen's flow law even though lab experiments indicate that 30 the flow law for isotropic ice really should depend on the third invariant of the deviatoric stress tensor (Morland, 2007, see e.g.). Sometimes tradition and convenience is a good explanation for why a particular simple model is used.

We acknowledge that additional work is needed to understand why ice has Schmidt-Ishlinsky like behaviour. But for context note that other polycrystalline materials, such as mild steel, copper, nickel alloy, titanium, stainless steel, have been found to have Schmidt-Ishlinsky like behaviour (e.g. Kolupaev, 2018, table 2.1).

We intend to make the following changes to the manuscript to address this comment:

– We will compare to additional criteria proposed in the literature. Specifically Tresca/Mohr-Coulomb. We can only compare to pressure independent criteria, as our method only can derive the deviatoric stress tensor. We intend to do this in the text using RMS misfit scores (see our response to comment 1).

– We will list other polycrystalline materials have been found to have Schmidt-Ishlinsky behaviour for context.

35

**Comment 3**

In the absence of this comparison, a deeper theoretical concern I have is that the S-I criterion doesn't appear to improve upon a key weakness of the vM criterion, which is that it is relatively insensitive to the direction of stress - indeed, it is noted by the authors that the difference between compressive and tensile ice strength is an order of magnitude, and that this isn't consistent with the vM criterion (L38-40). As the authors note, the S-I criterion does not significantly deviate strongly from this assumption (L154). Although they highlight that the S-I criterion implies ice is 15% stronger in shear relative to the tensile stress (L142), as far as I can tell from Fig. 4 the S-I criterion implies that ice is *exactly as strong* in tension and compression. However, examining Fig 4 appears to show far fewer crevasses in the compressive sections of the figure - especially if the color scale is log, which these plots often are. Therefore, although I am excited by the datasets and study design presented by the authors, the paper (at least, in its current form) leaves me questioning that crevasse failure can be adequately described by any prescribed radius in the $\pi$-plane.

**Response:** Thank you for the comment.

Yes – the S-I criterion is by definition equally strong in tensile vs compressive regimes. However, this is supported by the data. We calculated the empirical failure envelope (solid black line in fig4), and we can therefore determine the failure strength
40 of the tensile vs compressive directions directly from the data. This shows that the tensile failure strength is only 4% greater than the compressive failure strength (see figure 4). So, while this may be surprising, the data show that ice is nearly equally strong for tensile vs compressive deformation. This is part of the reason why the S-I criterion is a really good fit. We would therefore also disagree that this is a great weakness of the vM criterion.

Please note: The frequency of crevasses under different modes of deformation cannot be directly inform on the relative
45 strengths for each mode. For example, we see most crevasses in shear zones. That is probably simply because the greatest surface stresses produced by ice flow tend to occur in shear zones. The high frequency of shear crevasses **does not** mean that ice is weaker for shear (indeed we find the opposite to be the case). Similarly, we caution against interpreting the *"far fewer crevasses in the compressive sections"* in terms of a failure strength.

Finally, we do not understand the comment *"leaves me questioning that crevasse failure can be adequately described by any*
50 *prescribed radius in the $\pi$-plane."* The $\pi$-plane is just a pressure independent visualization method. **All** pressure independent failure criteria can be represented as a radius in the $\pi$-plane. The sum of the three principal deviatoric stresses is always zero, and so we effectively only have 2 degrees of freedom. The conclusions does not depend on the particular 2d projection we used. They would be the same if we had used a Vaughan (1993)-style projection.

We intend to make the following changes during revision:

- – We will add a discussion of the frequency distribution for different modes.

- – We will highlight the 4% empirical difference between Tensile and Compressive directions.

- – We will specify in the figure caption that the color scale is linear.

**55 Minor comments**

**Comment 4**

L95/Fig 1 - It is not mentioned in the methods exactly how the high-elevation exclusion zone is exactly derived. Manually determined? If so, the mask could be included as supplementary data.

**Response:** It is a manually traced polygon. We initially used a mask based on whether the elevation was above a sloped plane, but deemed it was just as arbitrary and required an overly elaborate explanation.

We intend to include the mask file as a data-supplement or archive the data in data repository.

**Comment 5**

L46 - Chudley et al. (2021) also use this data to assess crevasse formation, which is probably worth including/contrasting/comparing in the discussion.

60

**Response:** Thank you for the comment.
Agreed.

We intend to add a comparison to Chudley et al. (2021) failure envelopes.

**Comment 6**

L80/Fig 4 - Although I understand that plotting on the $pi$-plane is a key point of this paper, I imagine most will be more familiar with plotting on a simple $\tau_1/\tau_2$ plot following Vaughan (1993). I highly suggest including this alternative visualisation in the supplementary material to aid the interested reader in comparing and contrasting, as well as in understanding how this visualisation differs from Vaughan's approach.

65    **Response:** Thank you for the comment.

There are many ways to visualize the failure stress state (Kolupaev, 2018, ch3). While the style used by Vaughan (1993) is indeed widely used, the $\pi$-plane visualization is common in the broader material science community. We prefer following the material science community in this regard, as the $\pi$-plane plot is independent of pressure (i.e. plots for $\sigma$ or $\tau$ gives the same result) and takes all stress components into consideration. The style used by Vaughan (1993) is a particular 2D-slice through

70    the 3D stress failure space for $\sigma_3 = 0$ (Kolupaev, 2018, ch3.4), which makes the stress-state assumptions made not as clear as they could be. Several glaciological studies simply plot $\tau_1$ vs $\tau_2$ calculated from horizontal velocities, which we believe to be wrong or misleading; this suggests $\tau_3 = 0$, which is only true if there is zero horizontal divergence. If $\tau_3 = 0$ is not fulfilled, then we should no longer expect the points to fall on e.g. a vM ellipse, even if the points are generated by vM failure.

> We will consider adding a second panel to figure 4 with $\sigma_1$ vs $\sigma_2$ given $\sigma_3 = 0$ projection. I.e. plotting $\tau_1 - \tau_3$ against $\tau_2 - \tau_3$.

**Comment 7**

Fig 3 - Some indicator of y axis scale might be nice (unless normalized?)

75

**Response:** Thank you for the comment.
These are basically histograms normalized to have peak height 1.

> We intend to describe this in the caption.

**Comment 8**

Fig 4 - color scale needed for quantities.

80    **Response:** Thank you for the comment.
This is a 2D hexbin histogram. The gray color scale is linear in the counts per bin.

> We intend to describe the linear color scale in the caption.

> **Comment 9**
>
> L114-115 - Observational evidence of this can be found in recent papers (Harrington et al. 2017, doi:10.3189/2015AoG70A945; Hubbard et al. 2021, doi:10.1029/2020AV000291). I agree that it is likely that modelled MAT represents a lower bound of likely temperatures. Though for practical purposes, I don't have a better suggestion of how this can be approached.

**Response:** This is an important caveat as discussed in the manuscript. We also did not have a practical way of adjusting the temperature that we were happy with, and so we opted for the simplest solution where we use unadjusted CARRA along with a simple sensitivity calculation.

> In our revisions we will check if we can strengthen our discussion of this caveat using the references provided by the reviewer.

> **Comment 10**
>
> L123-128 - This is absolutely fascinating. How could the seasonally varying regions be better represented? Is it a case of crevasse initiation being initiated at the maximum velocity/stress? Or limited by the minimum velocity/stress?

**Response:** Our interpretation is that crevasses are initiated at the maximum velocity/stress.

It would be great if we could drop our constraint that limits our study to regions with steady ice flow. Ideally we would like high quality concurrent snapshot observations of crevassing and strain rates. Then we could directly relate the formation of new crevasses to changes in ice flow. However, such remote sensing products do not exist yet. Ice sheet wide "snapshot" velocity products, such as those provided by PROMICE, unfortunately have quite high levels of noise which in turn result in very noisy strain rates. This is why we had to use long-term average velocities for our analysis. We speculate that rather than post processing velocities, then it may be possible to make a new low-noise remotely sensed strain rate product using InSAR techniques (Andersen et al., 2020, see e.g.). Further, there is also no off-the-shelf product that reliably detects new crevasses over time for the entire ice sheet. But the rate of improvement in remote sensing products has been amazing, so we look forward to what the future will bring.

> We will verify that this is discussed in the manuscript. We believe it to be, but we may elaborate on this point.

> **Comment 11**
>
> L129-132 - Or limited by resolution/ability of crevasse dataset? The crevasse dataset is taken from another source and no limitations are discussed in the paper.

**Response:** We acknowledge that this might be an alternative explanation in some regions.

We intend to make the following revisions to address this comment:

– In these sentences: Add that another explanation might be limitations of crevasse data set.

– Describe the data limitations in the data section too.

**Comment 12**

L136 - does the data have a hexagonal pattern, or is the data clustered around the shear components and the hexagonal plotting style gives the impression that this is the case?

**Response:** The empirical failure envelope has a hexagonal pattern (Solid black line in fig. 4; see our response to review comment 1). This is not a feature of the plotting style.

We have outlined the changes we intend to make in order to address this point in our response to review comment 1.

**References**

Andersen, J. K., Kusk, A., Boncori, J. P. M., Hvidberg, C. S., and Grinsted, A.: Improved Ice Velocity Measurements with Sentinel-1 TOPS Interferometry, Remote Sensing, 12, 2014, https://doi.org/10.3390/rs12122014, 2020.

110    Chudley, T. R., Christoffersen, P., Doyle, S. H., Dowling, T. P. F., Law, R., Schoonman, C. M., Bougamont, M., and Hubbard, B.: Controls on Water Storage and Drainage in Crevasses on the Greenland Ice Sheet, Journal of Geophysical Research: Earth Surface, 126, e2021JF006 287, https://doi.org/10.1029/2021JF006287, 2021.

Kolupaev, V. A.: Equivalent Stress Concept for Limit State Analysis, vol. 86 of *Advanced Structured Materials*, Springer International Publishing, Cham, https://doi.org/10.1007/978-3-319-73049-3, 2018.

115    Morland, L. W.: The Influence of Third Shear Stress Invariant Dependence in the Isotropic Viscous Relation on the Reduced Model for Ice-Sheet Flow, Journal of Glaciology, 53, 597–602, https://doi.org/10.3189/002214307784409289, 2007.

Vaughan, D. G.: Relating the Occurrence of Crevasses to Surface Strain Rates, Journal of Glaciology, 39, 255–266, https://doi.org/10.3189/S0022143000015926, 1993.

---

## Author Response (AR1)

**Failure strength of glacier ice inferred from Greenland crevasses**

Addressed Comments for Publication to
EGUSphere (The Cryosphere)
by
A. Grinsted, N. Rathmann, R. Mottram, A. M. Solgaard, J. Mathiesen, and C.S. Hvidberg

**Authors' Response to the Editor**

> **General Comments.** Dear Aslak and co-authors,
>
> Thank you for your detailed and considered response to reviewer's comments on your manuscript. I now invite you to submit a revised manuscript which addresses these comments, paying particular attention to ensuring the choice and applicability of the S-I criterion is explained adequately.
>
> Best wishes, Caroline

**Response:** Dear Caroline

We appreciate your handling of our work.

We have addressed the reviewers comments as detailed below. Our 'author comments' to RC1 and RC2 had a list of changes we intended to make in this revision. This document is essentially a copy/paste of that, but where we outline the changes we actually ended up making in response to every comment.

> We have formatted our accounting of changes like this.

In particular, we have made many revisions to the 'failure criterion' subsection to ensure that we explain the choice and applicability of the S-I criterion adequately (as detailed below).

Thank you, Aslak and co-authors

**Response to RC1**

> **General Comments.** Grinsted et al. use ice-sheet-wide geophysical datasets to assess the applicability of a previously untested failure criterion, the Schmidt-Ishlinsky (hereafter S-I) criterion, to predict ice failure across the Greenland Ice Sheet. This is done with the intention of being able to develop accurate predictors of crevasse formation and presence in large-scale ice-flow and fracture models. The paper is timely and is a clear evolution of the ongoing large-scale work being done on ice failure in recent years. It is a bit of an "open secret" that von Mises (hereafter vM) is a poor predictor of crevasse failure and it is good to see work being done to find better approaches, especially involving large-scale datasets. I have some thoughts regarding the failure criterion chosen, and in particular the lack of wider assessment given the prime opportunity provided by the datasets collated.

**Response:** Thank you for your feedback.

From our perspective we would not say that we assess the applicability of the Schmidt-Ishlinsky criterion, as it is not a criterion we propose a priori. Rather we make an empirical study of the failure criterion, and then find that it happens to be well modelled by the S-I criterion.

We hope that we address your detailed comments in our replies below.

> **Comment 1**
>
> As written, it is unclear what motivation or hypothesis led the authors to propose and test the S-I failure criterion. After the abstract, the criterion is not mentioned again until the discussion (L136), where it is defined but without any clear motivation as to why. At this point, it is argued that although both vM and S-I perform quantitatively identically (L144), if measurement noise was better S-I would likely be the better performer (L144-146). It is not possible for the reader to assess whether S-I performs better or worse than any other alternative criteria common in the literature, as this data is 'not shown' (L147-148).

25

**Response:** The motivation for the S-I criterion is purely empirical - It is *the only* simple failure criteria that fits the data (simple in the sense that it has no shape parameters). We realize that this may not have been clear as S-I is introduced in the methods before any results are shown.

Here's how we arrived at that conclusion. We calculate an empirical failure envelope by calculating the moving $5°$ median

30 radial distance in the $\pi$-plane. This is shown as a fat solid black line in figure 4. It has a distinct hexagonal shape that we can compare that against a library of failure envelopes (see fig3.6 and fig3.7 in Kolupaev, 2018). We judged that this was so distinctly Schmidt-Ishlinsky type that we not need to quantitatively assess the misfit of different criteria. However, we have since done so, and find that the root mean square log misfits of the S-I, vM, and Tresca envelopes are 0.02, 0.04 and 0.08 respectively. Hence, we can now also conclude that the S-I criterion objectively provides the best fits to the data.

> We have made the following changes to address this comment:
>
> – We now clearly state that we arrive at the S-I criterion by comparing the empirical envelope against a library of common criteria.
>
>> "However, the data clearly has a hexagonal pattern which we can compare against a library of common failure criteria (Kolupaev, 2018, ch.3). The hexagonal shape, with corners in the shear directions, strongly suggests that glacier ice follows a Schmidt–Ishlinsky failure criterion (Burzynski, 1928; Schmidt, 1932; Yu, 1983; Kolupaev, 2018)."
>
> – Expanded on the description of how we calculated the empirical failure envelope.
>
> – Score the different failure criteria in terms of a RMS misfit between the proposed failure envelopes and the empirical failure envelope.
>
> – Include the misfit score of additional (worse) failure criteria in the literature such as the Tresca/Mohr-Coulomb.
>
> – Argued in more detail for why we can reject the maximum strain energy criterion.
>
> These changes are primarily in the 'Failure criterion' subsection of the manuscript.

35

**Comment 2**

The S-I is not a criterion I have encountered before in the glaciological literature, and appears to be pretty niche outside the discipline as well - indeed, googling the phrase 'Schmidt-Ishlinsky failure criterion' includes this preprint among the top results. As a result, I think it is important that the authors do more to contextualise and explain the criterion, and their motivation for choosing it. This is especially true as the authors disregard even comparing the criterion to other options, as previously noted (L147-148). Given the clear effort that has been put into producing and collating the various ice-sheet-wide datasets, it would be very interesting to see a comparison/EDA of all previously suggested criteria, and make it more convincing that the S-I criterion is (qualitatively) a better option.

**Response:** The motivation for the criterion is entirely empirical. It simply fits the data best (see our response to comment 1).

The difference between the vM criterion and S-I is relatively subtle considering the imperfections of real world data. So, it is not that surprising to us that the S-I like behaviour of glacier ice has not been spotted before. It may seem like an exotic criterion, but consider that nearly the entire ice flow literature uses Glen's flow law even though lab experiments indicate that the flow law for isotropic ice really should depend on the third invariant of the deviatoric stress tensor (Morland, 2007, see e.g.). Sometimes tradition and convenience is a good explanation for why a particular simple model is used.

We acknowledge that additional work is needed to understand why ice has Schmidt-Ishlinsky like behaviour. But for context note that other polycrystalline materials, such as mild steel, copper, nickel alloy, titanium, stainless steel, have been found to have Schmidt-Ishlinsky like behaviour (e.g. Kolupaev, 2018, table 2.1).

We have made the following changes to address this comment:

  – We have compared to additional criteria proposed in the literature. Specifically Tresca/Mohr-Coulomb, von Mises, and the maximum strain energy. (see also our response to comment 1).

  – We have listed other polycrystalline materials with Schmidt-Ishlinsky behaviour for context.

**Comment 3**

In the absence of this comparison, a deeper theoretical concern I have is that the S-I criterion doesn't appear to improve upon a key weakness of the vM criterion, which is that it is relatively insensitive to the direction of stress - indeed, it is noted by the authors that the difference between compressive and tensile ice strength is an order of magnitude, and that this isn't consistent with the vM criterion (L38-40). As the authors note, the S-I criterion does not significantly deviate strongly from this assumption (L154). Although they highlight that the S-I criterion implies ice is 15% stronger in shear relative to the tensile stress (L142), as far as I can tell from Fig. 4 the S-I criterion implies that ice is *exactly as strong* in tension and compression. However, examining Fig 4 appears to show far fewer crevasses in the compressive sections of the figure - especially if the color scale is log, which these plots often are. Therefore, although I am excited by the datasets and study design presented by the authors, the paper (at least, in its current form) leaves me questioning that crevasse failure can be adequately described by any prescribed radius in the $\pi$-plane.

**Response:** Thank you for the comment.

Yes – the S-I criterion is by definition equally strong in tensile vs compressive regimes. However, this is supported by the data. We calculated the empirical failure envelope (solid black line in fig4), and we can therefore determine the failure strength of the tensile vs compressive directions directly from the data. This shows that the tensile failure strength is only 4% greater than the compressive failure strength (see figure 4). So, while this may be surprising, the data show that ice is nearly equally

strong for tensile vs compressive deformation. This is part of the reason why the S-I criterion is a really good fit. We would therefore also disagree that this is a great weakness of the vM criterion.

Please note: The frequency of crevasses under different modes of deformation cannot be directly inform on the relative strengths for each mode. For example, we see most crevasses in shear zones. That is probably simply because the greatest surface stresses produced by ice flow tend to occur in shear zones. The high frequency of shear crevasses **does not** mean that ice is weaker for shear (indeed we find the opposite to be the case). Similarly, we caution against interpreting the *"far fewer crevasses in the compressive sections"* in terms of a failure strength.

Finally, we do not understand the comment *"leaves me questioning that crevasse failure can be adequately described by any prescribed radius in the $\pi$-plane."* The $\pi$-plane is just a pressure independent visualization method. **All** pressure independent failure criteria can be represented as a radius in the $\pi$-plane. The sum of the three principal deviatoric stresses is always zero, and so we effectively only have 2 degrees of freedom. The conclusions does not depend on the particular 2d projection we used. They would be the same if we had used a Vaughan (1993)-style projection.

> We have made the following changes during revision:
>
> – We have included a brief discussion of the frequency distribution for different modes to the 'Failure criterion' subsection.
>
> – We have highlighted the 4% empirical difference between Tensile and Compressive directions in the 'Failure criterion' subsection.
>
> – The figure caption now states that the color scale is linear in the counts per bin.

**Minor comments**

> ### Comment 4
>
> L95/Fig 1 - It is not mentioned in the methods exactly how the high-elevation exclusion zone is exactly derived. Manually determined? If so, the mask could be included as supplementary data.

**Response:** It is a manually traced polygon. We initially used a mask based on whether the elevation was above a sloped plane, but deemed it was just as arbitrary and required an overly elaborate explanation.

> We have published the mask and some additional supplemental data files on zenodo here: (Grinsted, 2024).
> Changes made: added "using a manually traced polygon" to the text.

> ### Comment 5
>
> L46 - Chudley et al. (2021) also use this data to assess crevasse formation, which is probably worth including/contrasting/comparing in the discussion.

**Response:** Thank you for the comment.
Agreed.

> We have revised the discussion to include the following:
>
> > "We interpret this as our method systematically underestimating the failure stress in regions with strong seasonal variations in near surface strain rates, and therefore argue that onset regions with steady flow more accurately represent the failure stress. Excluding regions with non-steady flow result in a tighter stress distribution that has an improved separation from the stress distribution over ice sheet as a whole (fig. 3). We therefore estimate the failure stress to be $\tau_{\mathrm{vM}} = (265 \pm 73)\,\mathrm{kPa}$ from onset regions with steady flow. This separation is not as clear in Chudley et al. (2021) which is based on a regional analysis of the same crevasse dataset but does not exclude areas with non-steady flow."
>
> We do not speculate further on why the Chudley et al. (2021) failure map analysis is unable to clearly separate stresses in initiating crevasse and non-crevassed regions. We can think of many potential reasons:
>
> – They may underestimate peak-stress because they use average velocities to derive strain rates. This is why we end up excluding regions with non-steady flow in our study. –as alluded to in our revised discussion.
>
> – They do not take incompressibility and the vertical deviatoric stress into account.
>
> – They plot horizontal *deviatoric* stresses against each other in their failure maps (instead of the stresses). This is not strictly correct, as you really should plot stresses in this type of plot (Kolupaev, 2018, ch. 3.4). This gives misleading results, that are easy to misinterpret. We believe that this is one reason for why their data appear to be less elongated along the $x = y$ axis than you would expect from e.g. von Mises failure (Chudley et al., 2021, fig 5). Brief explanation: consider the case where $\tau_1 = \tau_2 = 1$. In that case, $\tau_3 = -2$, by definition. If we further impose $\sigma_3 = 0$, then this implies that we should plot $\sigma_1 = \tau_1 - \tau_3$ against $\sigma_2 = \tau_2 - \tau_3$. I.e. rather than plotting at the coordinate $(1,1)$, then we should plot at $(3,3)$.
>
> – They used different criteria for deciding if we are in an crevasse onset/crevasse initiating region.
>
> – They use a single temperature everywhere. Any local temperature deviations over the region of interest will 'smear' the inferred stresses. This could be important as they consider a region with a rather large elevation span.
* * *
**Comment 6**

L80/Fig 4 - Although I understand that plotting on the $pi$-plane is a key point of this paper, I imagine most will be more familiar with plotting on a simple $\tau_1/\tau_2$ plot following Vaughan (1993). I highly suggest including this alternative visualisation in the supplementary material to aid the interested reader in comparing and contrasting, as well as in understanding how this visualisation differs from Vaughan's approach.

**Response:** Thank you for the comment.
There are many ways to visualize the failure stress state (Kolupaev, 2018, ch3). While the style used by Vaughan (1993) is indeed widely used, the $\pi$-plane visualization is common in the broader material science community. We prefer following the material science community in this regard, as the $\pi$-plane plot is independent of pressure (i.e. plots for $\sigma$ or $\tau$ gives the same result) and takes all stress components into consideration. The style used by Vaughan (1993) is a particular 2D-slice through the 3D stress failure space for $\sigma_3 = 0$ (Kolupaev, 2018, ch3.4), which makes the stress-state assumptions made not as clear as

they could be. Several glaciological studies simply plot $\tau_1$ vs $\tau_2$ calculated from horizontal velocities, which we believe to be wrong or misleading; this suggests $\tau_3 = 0$, which is only true if there is zero horizontal divergence. If $\tau_3 = 0$ is not fulfilled, then we should no longer expect the points to fall on e.g. a vM ellipse, even if the points are generated by vM failure.

> We have elaborated on why we prefer the $\pi$-plane visualization in the "$\pi$-plane" subsection.

85

**Comment 7**

Fig 3 - Some indicator of y axis scale might be nice (unless normalized?)

**Response:** Thank you for the comment.
These are basically histograms normalized to have peak height 1.

> We have describe this better in the caption.

**Comment 8**

Fig 4 - color scale needed for quantities.

90

**Response:** Thank you for the comment.
This is a 2D hexbin histogram. The gray color scale is linear in the counts per bin.

> We have revised the caption to include a description of the linear color scale.

**Comment 9**

L114-115 - Observational evidence of this can be found in recent papers (Harrington et al. 2017, doi:10.3189/2015AoG70A945; Hubbard et al. 2021, doi:10.1029/2020AV000291). I agree that it is likely that modelled MAT represents a lower bound of likely temperatures. Though for practical purposes, I don't have a better suggestion of how this can be approached.

95   **Response:** This is an important caveat as discussed in the manuscript. We also did not have a practical way of adjusting the temperature that we were happy with, and so we opted for the simplest solution where we use unadjusted CARRA along with a simple sensitivity calculation.

The manuscript already discuss the potential temperature bias caveat sufficiently, and we have therefore not added any additional text to these paragraphs. We have added the two papers as additional citations with examples of how temperatures can deviate strongly from surface temperatures. The two papers are however not directly relevant to "crevasse onset" with "steady flow" regions that are the focus of our study:

- Harrington et al. (2015) shows large temperature anomalies at the depths we are interested in. However, these are interpreted as being due to advection of melt water filled crevasses. However, we focus on crevasse **onset** where there should no/few upstream crevasses.

- Hubbard et al. (2021) finds refrozen melt water layers in what they interpret to be previously open crevasses that are now 'healed'. I.e. advected crevasses or crevassing from non-steady flow. We explicitly exclude regions downstream from crevasse onset, and regions with non-steady flow.

**Comment 10**

L123-128 - This is absolutely fascinating. How could the seasonally varying regions be better represented? Is it a case of crevasse initiation being initiated at the maximum velocity/stress? Or limited by the minimum velocity/stress?

100    **Response:** Our interpretation is that crevasses are initiated at the maximum velocity/stress.

It would be great if we could drop our constraint that limits our study to regions with steady ice flow. Ideally we would like high quality concurrent snapshot observations of crevassing and strain rates. Then we could directly relate the formation of new crevasses to changes in ice flow. However, such remote sensing products do not exist yet. Ice sheet wide "snapshot" velocity products, such as those provided by PROMICE, unfortunately have quite high levels of noise which in turn result in

105    very noisy strain rates. This is why we had to use long-term average velocities for our analysis. We speculate that rather than post processing velocities, then it may be possible to make a new low-noise remotely sensed strain rate product using InSAR techniques (Andersen et al., 2020, see e.g.). Further, there is also no off-the-shelf product that reliably detects new crevasses over time for the entire ice sheet. But the rate of improvement in remote sensing products has been amazing, so we look forward to what the future will bring.

The revisions introduced to address review comment 5 clarifies these sentences. No other changes were made.

110

**Comment 11**

L129-132 - Or limited by resolution/ability of crevasse dataset? The crevasse dataset is taken from another source and no limitations are discussed in the paper.

**Response:** We acknowledge that this might be an alternative explanation in some regions.

We have made the following revisions to address this comment:

  – Added that another explanation might be limitations of crevasse data set.

  – Describe data limitations (in particular that it ArcticDEM derived dataset will be insensitive to snow-filled crevasses. This has been added to the data section.

**Comment 12**

L136 - does the data have a hexagonal pattern, or is the data clustered around the shear components and the hexagonal plotting style gives the impression that this is the case?

115    **Response:** The empirical failure envelope has a hexagonal pattern (Solid black line in fig. 4; see our response to review comment 1). This is not a feature of the plotting style.

The entire subsection has been revised substantially for clarity. We have outlined the changes made in our response to review comment 1.

**Response to RC2**

**General Comments.** This is an excellent paper, addressing an under-studied but important issue in glaciology in an elegant and rigorous manner. The methods and results are presented efficiently and clearly, with enough detail to address the important issues but without clutter or un-necessary material. It is pleasing to see such clear patterns emerge from the data, despite the many potential issues with data resolution. My only substantial criticism is that the crevasse onset criteria should also be presented in terms of strain rates, rather than stress metrics alone. As the authors clearly state, the calculated stresses depend on the choice of rheology. Standard values have been used, and the prefactor A has been scaled to temperature; this is all good, and aligns with standard practice in glaciology. However, major sources of uncertainty remain, including the true temperatures at crevasse-initiation depth, non-temperature influences on A, and the possibility (indeed, likelihood) of varying $n$ across the very large study area. For these reasons, the calculated stresses are not absolute, but parameter-dependent. The authors have done an excellent job of highlighting these issues in the text, and I have no issue with that. However, it would be very useful to present the raw strain rate values – these are the observations, and are hence free from any assumptions regarding the rheology. Including the strain rate data will offer researchers greater flexibility in how they interpret and use the results presented in this paper. I do not see any need to adjust what is already written in the paper, simply to add a section (and a Figure) on the strain rates.

I found only one typo. On line 82, one 'principle' sneaked into the text. As is the case elsewhere in the paper, this should of course be 'principal'.

The author team are to be congratulated on a fine study. The paper is likely to be widely cited: I shall certainly find it very useful for my work on the role of crevasses in calving.

Doug Benn

120    **Response:** Thank you for the kind words. We hope our work will be useful.

We have made a preliminary attempt at a strain rate figure as requested (fig. R1). It is certainly interesting. However, we are not entirely happy with how it has turned out. 1) The 2d histogram below the plot is nearly invisible because of the very long tailed strain rate distribution. 2) It is very dependent on temperature which makes it difficult to choose an appropriate "zoom" level.

[Figure]

**Figure R1.** Empirical failure map showing a $\pi$-plane density map of strain rates in crevasse onset regions with steady flow. The empirical median in $10°$ windows for three different temperatures is shown as fat lines; Tensile, Compressive and Shear directions have been labelled with T, C, and S.
* * *
The following changes were made to the manuscript during revision:

– Typo fixed.

125    – The strain rate 'flower' figure, and a brief discussion of it, has been added to the appendix.
* * *
**References**

Andersen, J. K., Kusk, A., Boncori, J. P. M., Hvidberg, C. S., and Grinsted, A.: Improved Ice Velocity Measurements with Sentinel-1 TOPS Interferometry, Remote Sensing, 12, 2014, https://doi.org/10.3390/rs12122014, 2020.

Burzynski, W.: Studjum Nad Hipotezami Wytężenia, Nakladem Akademii nauk Technicznych, 1928.

Chudley, T. R., Christoffersen, P., Doyle, S. H., Dowling, T. P. F., Law, R., Schoonman, C. M., Bougamont, M., and Hubbard, B.: Controls on Water Storage and Drainage in Crevasses on the Greenland Ice Sheet, Journal of Geophysical Research: Earth Surface, 126, e2021JF006 287, https://doi.org/10.1029/2021JF006287, 2021.

Grinsted, A.: Supplemental Data for Failure Strength of Glacier Ice Inferred from Greenland Crevasses, https://doi.org/10.5281/zenodo.10567694, 2024.

Harrington, J. A., Humphrey, N. F., and Harper, J. T.: Temperature Distribution and Thermal Anomalies along a Flowline of the Greenland Ice Sheet, Annals of Glaciology, 56, 98–104, https://doi.org/10.3189/2015AoG70A945, 2015.

Hubbard, B., Christoffersen, P., Doyle, S. H., Chudley, T. R., Schoonman, C. M., Law, R., and Bougamont, M.: Borehole-Based Characterization of Deep Mixed-Mode Crevasses at a Greenlandic Outlet Glacier, AGU Advances, 2, e2020AV000 291, https://doi.org/10.1029/2020AV000291, 2021.

Kolupaev, V. A.: Equivalent Stress Concept for Limit State Analysis, vol. 86 of *Advanced Structured Materials*, Springer International Publishing, Cham, https://doi.org/10.1007/978-3-319-73049-3, 2018.

Morland, L. W.: The Influence of Third Shear Stress Invariant Dependence in the Isotropic Viscous Relation on the Reduced Model for Ice-Sheet Flow, Journal of Glaciology, 53, 597–602, https://doi.org/10.3189/002214307784409289, 2007.

Schmidt, R.: Über den Zusammenhang von Spannungen und Formänderungen im Verfestigungsgebiet, Ingenieur-Archiv, 3, 215–235, https://doi.org/10.1007/BF02079970, 1932.

Vaughan, D. G.: Relating the Occurrence of Crevasses to Surface Strain Rates, Journal of Glaciology, 39, 255–266, https://doi.org/10.3189/S0022143000015926, 1993.

Yu, M.-h.: Twin Shear Stress Yield Criterion, International Journal of Mechanical Sciences, 25, 71–74, https://doi.org/10.1016/0020-7403(83)90088-7, 1983.